# Socioeconomic inequalities in utilizing facility delivery in Bangladesh: A decomposition analysis using nationwide 2017–2018 demographic and health survey data

Md. Ashfikur Rahman[1]☯*, Satyajit Kundu[2,3,4]☯, Harun Or Rashid[5], Hasibul Hasan Shanto[5], Md. Mahmudur Rahman[6], Bayezid Khan[1], Md. Hasan Howlader[1], Md. Akhtarul Islam[5]

1 Development Studies Discipline, Social Science School, Khulna University, Khulna, Bangladesh, 2 Global Health Institute, North South University, Dhaka, Bangladesh, 3 School of Public Health, Southeast University, Nanjing, China, 4 Faculty of Nutrition and Food Science, Patuakhali Science and Technology University, Patuakhali, Bangladesh, 5 Statistics Discipline, Khulna University, Khulna, Bangladesh, 6 Statistics Department, University of Rajshahi, Rajshahi, Bangladesh

☯ These authors contributed equally to this work.
* ashfikur@ku.ac.bd

**Data Availability Statement:** The data underlying the results presented in the study are available from Demographic Health Survey (https://

## Abstract

### Background

In many low- and middle-income countries (LMICs), including Bangladesh, socioeconomic inequalities in access to maternity care remain a substantial public health concern. Due to the paucity of research, we attempted to determine the factors affecting the facility delivery, quantify wealth-related inequality, and identify potential components that could explain the inequality.

### Methods

We used the latest Bangladesh Demographic and Health Survey (BDHS 2017–18) data in this study. We utilized logistic regression to investigate the associated factors of facility delivery. The concentration curves (CC), concentration index (CIX) and decomposition of CIX techniques were used to analyze the inequality in-facility delivery.

### Results

Women living in the urban areas, age at first birth after (18–24 years ≥25 years), being over-weight/obese, having secondary and higher-level education of the women and their husband, seeking four or more ANC, coming from more affluent households, and women with high enlightenment were significant determinants of facility delivery. The concentration curve was below the line of equality, and the relative concentration index (CIX) was 0.205 (p <0.001), indicating that women from wealthy groups were disproportionately more prevalent to facility delivery. The decomposition analysis reveals that wealth status of women (57.40%), age at first birth (10.24%), husband's education (8.96%), husband's occupation

dhsprogram.com/) upon reasonable request to the DHS platform which is free of cost.

**Funding:** The author(s) received no specific funding for this work.

**Competing interests:** The authors have declared that no competing interests exist.

(7.35%), education of women (7.13%), women's enlightenment (6.15%), residence (8.64%) and ANC visit (6.84%) are the most major contributors to the inequalities in utilizing facility delivery.

## Conclusion

The study demonstrates a clear disparity in the use of facility delivery among Bangladeshi women; hence, immediate action is required to lower the inequalities, with a special emphasis on the contributing factors.

## Background

Every day, an estimated 810 maternal deaths occur worldwide, with 94% of these deaths occurring in low- and -middle-income nations (LMICs), despite the fact that these deaths are avoidable. The global maternal mortality ratio (MMR or the number of maternal deaths per 100,000 live births) decreased by approximately 38% between 2000 and 2017 [1]. South Asia and Sub-Saharan Africa accounted for 86% (254,000) of all maternal deaths in 2017, with South Asia accounting for almost one-fifth of all deaths (58,000). However, since 2000, remarkable improvement has been made. South Asia's MMR decreased by 59% between 2000 and 2017 (from 395 to 163 maternal deaths per 100,000 live births), while Sub-Saharan Africa's MMR decreased by 39% within the same time frame [2,3].

The Millennium Development Goals (MDGs) 1990–2015 emphasized the critical need to halve infant and maternal mortality. This has significantly decreased the global maternal mortality rate (MMR) to 38% by 2015 [1]. SDG 3 targets to reduce MMR mortality to less than 70 per 100,000 live births by 2030. Setting this high goal for SDGs is a good rallying cry for all nations, but the objectives from the MDG era are still not complete. Such progress made during the era of MDGs has not been distributed equitably throughout the world. In recent decades, Bangladesh, Nepal, and Pakistan have made commendable strides in reducing MMR. Between 2010 and 2017, the MMR decreased to 173/100,000 live births in Bangladesh, 186/100000 in Nepal, and 140/1000000 in Pakistan [4]. Nevertheless, MMR rates in these nations remain extremely higher than in other LMICs worldwide.

In 2017, 295,000 women died because of complications related to pregnancy, low rates of facility-based delivery and a lack of skilled birth attendants (SBAs) during pregnancy or childbirth. These are the key reasons for high maternal death rates in these regions which could possibly be averted by shifting birth to a health facility [5–7]. According to previous studies [8–11], the majority of maternal deaths occur when mothers do not receive professional treatment during home birth to handle haemorrhage, sepsis, botched abortion, obstructed labor, or hypertensive problems.

In addition, unsafe home delivery practices are the cause of 35% of all antepartum, intrapartum and postpartum hemorrhage [6,12,13]. A large number of women in LMICs countries continue to receive inadequate care throughout pregnancy and childbirth period [11] but 80% of MMR from LMICs can be avoided by utilizing SBA and facility delivery [8,11]. Family planning, prenatal, and postnatal care, as well as the use of facility-based delivery services may aid to the reduction of maternal mortality [12,14,15]. However, a variety of issues limit the use of facility delivery and associated services during the pregnancy to postnatal period. Poor health seeking behavior, inadequately developed health systems, low socioeconomic status, cultural and personal health beliefs, lack of appropriate health services, high cost, long distance of

service centers, lack of transportation facilities, and poor quality of treatments have been identified as the critical barriers to low health care utilization [8,16]. In Bangladesh, women who deliver at home are more likely to be affected by unhygienic environment, putting the lives of mothers and newborns at risk of dangerous conditions [11,13].

Investigating to what extent the socioeconomic inequalities exist in facility delivery might help in identifying the primary causes of these lower utilization and thus guide relevant stakeholders on how to alleviate these inequalities. Additionally, it is essential to examine the determinants of such inequality so that policymakers can develop evidence-based policy interventions. Few research analyzes the socioeconomic factors of inequalities in utilizing facility delivery among women in Bangladesh using a nationally representative sample from the 2017–2018 Bangladesh Demographic and Health Survey (BDHS) data. The principal objectives of this study are three-fold: (i) to analyze the factors of facility delivery in Bangladesh using the most recent round nationally representative survey; (ii) to measure the socio-economic inequality in the use of facility delivery; (iii) to identify the primary components that explain socioeconomic inequality in facility delivery through decomposition analysis.

## Methods

### Data sources

The study utilized secondary data from the most recent Demographic and Health Surveys (DHS) (BDHS 2017–18). Demographic and Health Surveys are periodic surveys conducted to ascertain the population's health status. A DHS survey provides a comprehensive picture of the study population including the overall status of mother and child health as well as a variety of other health care theme areas. The dataset has been kept free to access for the academics and researchers for their use from the internet. All protocols for DHS surveys were ethically approved by the Institutional Review Board and country-specific review bodies. The final report contains a full overview of the survey strategy, methodology, sampling, and questionnaires [17]. A total of 20,127 ever-married women aged between 15 and 49 years were interviewed out of 20,376 eligible women. Of them, 4,814 women were considered for final analysis since they met the inclusion criteria. We set the inclusion criteria those women had at least one birth, age 15 to 49 years and valid information on last delivery details.

### Outcome variable

Place of delivery (0 = Home, 1 = Facility) was the outcome variable in our analyses. The place of delivery was considered 'facility' if a woman gave their last birth in a government hospital, district hospital, maternal and child welfare center (MCWC), Upazila health complex, health and family welfare center, private hospital/clinic, private medical college/hospital, rural health center, basic health unit, primary health care center and outreach clinic, or in a clinic run by family planning association. It was considered 'home delivery' if a woman gave birth at the respondent's own or relative's/neighbor's home.

### Explanatory variables

A systematic literature search was performed to identify the predictor variables. A list of data (respondents' involvement in deciding on their healthcare, decision on large household purchases, and decision on visits to family or relatives) was used to measure respondents' decision-making power. The enlightenment level of mothers was measured using educational attainment, newspaper/magazines reading, radio listening, and television watching while wife beating was quantified by aggregating responses from women and categorizing them as low or

**Table 1. Description of the explanation variables along with their categories.**

| SL. No. | Variables | Construction |
|---|---|---|
| 1 | Place of Residence | Rural, Urban |
| 2 | Division | Barishal, Chattogram, Dhaka, Khulna, Mymensingh, Rajshahi, Rangpur, Sylhet |
| 3 | Age of the mother | 15–24, 25–34, 35–49 |
| 4 | Age at first Birth | <18, 18–24, ≥25 |
| 5 | Mother's BMI | Underweight, Normal, Overweight/Obese |
| 6 | Mother's Education | No education, Primary, Secondary, Higher |
| 7 | Mother's Employment status | Working, Not Working |
| 8 | Number of ANC Visits | Nil, 1–3, ≥4 |
| 9 | Decision-Making Power | Low, High |
| 10 | Mother's Enlightenment | Low, Medium, High |
| 11 | Wife Beating | Low, High |
| 12 | Husband's Education | No education, Primary, Secondary, Higher |
| 13 | Husband's Occupation | Agricultural, Business, Job/Services, Others |
| 14 | Household Wealth Status | Poorest, Poorer, Middle, Richer, Richest |

high. The following items were also used: "beating justified if wife goes out of home without telling husband", "beating justified if wife neglects the children", "beating justified if wife argues with husband", "beating justified if wife refuses to have sex with husband", and "beating justified if wife burns the food" for the analysis of the study. Using Principal Component Analysis (PCA), the factors were distilled into a more generalised set of weights that score between 0 and 100 to "women's enlightenment" and "decision making power". The standardized z-scores were used to differentiate between overall low, medium, and high scores [18–20]. The detailed description of the study variables has been discussed in the following (**Table 1**).

## Statistical analysis

The background characteristics of the study population were summarized using descriptive statistics while weighted prevalence with 95% CI was presented. Unadjusted regression tests were used to determine the relationship between the predictor variable and the delivery location. After controlling for confounding variables, multivariate logistic regression was used to determine the net influence of predictor variables on the outcome variable (facility delivery vs. home delivery), where only the significant (at <0.05 level) variables in the unadjusted model were included in the adjusted regression model. The findings section presents the factors that are statistically significant at 0.05 level in the adjusted model. This article presents both unadjusted/crude odds ratios (cOR) and adjusted odds ratios (AOR) but for results interpreting authors only used the adjusted model. The statistical program Stata/MP 16 (StataCorp, College Station, Texas, USA) was used to conduct all analyses.

## Inequality measurement

The concentration curve (CC) and concentration index (CIX) in their relative formulation (with no correction), were used to investigate the inequality in terms of utilizing facility delivery across analyzable socio-economic characteristics of the population (women) [21]. The CIX in this study represents horizontal inequity, as each woman in the study was presumed to have an equal requirement for facility delivery. While constructing CC, the cumulative proportion

of women ranked according to their wealth index score (poorest first) was plotted against the cumulative proportion of facility deliveries on the y-axis. The 45-degree slope from the origin revealed perfect equality. If the CC overlaps with the line of equality, utilization of institutional delivery is equal among women. However, if the CC subtends the line of equality below (above), then inequality in the use of institutional delivery exists and is slanted towards women belonging to low (high) socioeconomic background. The further the CC subtends from the line of equality, the greater the degree of inequality. To evaluate the extent of wealth-related inequality, CIX was determined. CIX is widened as twice the region between the line of equality and CC [21].

The CIX takes a value between −1 and +1. When the institutional delivery is equally distributed across socio-economic groups, CIX takes the value of 0. A positive value of CIX implies that the use of institutional delivery is concentrated among the higher socio-economic groups (pro-rich). When institutional delivery is distributed equally across socioeconomic classes, CIX equals zero. A positive number for CIX indicates that institutional delivery is more prevalent among the more affluent socioeconomic categories (pro-rich). In contrast, a negative value of CIX indicates that institutional delivery is primarily used by lower socioeconomic groups (pro-poor). The calculation of CIX was done by using "convenient covariance" formula described by O'Donnell et al. [21], as shown in the Eq 1 below.

$$CIX = \frac{2}{\mu} cov\ (h, r) \tag{1}$$

Here $h$ is the health sector variable, $\mu$ is its mean, and $r = i/N$ is the fractional rank of individual $i$ in the living standards distribution, with $i = 1$ for the poorest and $i = N$ for the richest. The user-written STATA commands Lorenz [22] and conindex [23] were used to produce CC and measure CIX, respectively.

## Decomposition of CIX

The relative CIX was decomposed to determine the portion of inequality owing to the inequality in the underlying determinants. The findings were analyzed and interpreted using the technique described by Wagstaff et al. and O'Donnell et al. The contribution of each determinant of facility delivery to overall wealth-related inequality is determined as the product of the determinant's sensitivity to facility delivery (elasticity) and the degree of wealth-related inequality in that determinant (CIX of determinant). The residual is the portion of the CIX that is not explained by the determinants.

## Results

Table 2 represents the distribution of utilizing facility births by household wealth quintile for women of reproductive aged women in Bangladesh. In the present study, the highest percentage of women who utilized facility delivery was recorded in Khulna (62.2%), followed by Dhaka (60.3%), and Sylhet (41.8%). It has also been observed that, for all categories, women with a secondary level education mostly utilize facility delivery (50.8%) while women with no education utilize the least amount of it. The only exceptions are higher educated women with age ≥25 years (65.5%), and whose husbands also had a higher level of education (63.2%). In addition, women who visited ANC (≥4) utilized more facility delivery (66.1%) than those who did not.

Table 3 represents the association between dependent and independent variables of interest. All of the independent variables, except the age of the mother, and her decision-making

**Table 2. Distribution of utilizing facility birth by analyzable reproductive aged women in Bangladesh: BDHS 2017–18.**

| | | Total (n/%) | Facility birth (%) | Poorest | Poorer | Middle | Richer | Richest |
|---|---|---|---|---|---|---|---|---|
| **Divisions** | | | | | | | | |
| | Barishal | 518 (10.8) | 41.9 | 8.5 | 5.4 | 6.2 | 3.9 | 2.1 |
| | Chattogram | 796 (16.5) | 46.5 | 10.7 | 13.1 | 21.8 | 20.9 | 23.4 |
| | Dhaka | 672 (14.0) | 60.3 | 11.9 | 13.7 | 22.1 | 32.6 | 44.3 |
| | Khulna | 495 (10.3) | 62.2 | 8.5 | 12.6 | 15.0 | 10.9 | 8.7 |
| | Mymensingh | 589 (12.2) | 42.4 | 9.6 | 12.1 | 6.6 | 6.3 | 3.6 |
| | Rajshahi | 497 (10.3) | 55.9 | 15.9 | 20.4 | 13.5 | 13.2 | 5.7 |
| | Rangpur | 550 (11.4) | 52.0 | 27.4 | 15.5 | 10.3 | 6.3 | 5.6 |
| | Sylhet | 697 (14.5) | 41.8 | 7.4 | 7.2 | 4.5 | 5.9 | 6.6 |
| **Place of Residence** | | | | | | | | |
| | Urban | 1610 (33.4) | 62.5 | 2.7 | 5.1 | 10.6 | 25.4 | 56.2 |
| | Rural | 3204 (66.6) | 43.6 | 15.4 | 20.6 | 23.8 | 24.2 | 16.0 |
| **Age of the mother** | | | | | | | | |
| | 15–24 | 2563 (53.2) | 51.1 | 64.3 | 63.2 | 54.4 | 55.9 | 47.9 |
| | 25–34 | 1974 (41.0) | 49.1 | 32.3 | 33.3 | 39.6 | 40.4 | 46.9 |
| | 35–49 | 277 (5.8) | 45.1 | 3.3 | 3.5 | 6.0 | 3.7 | 5.3 |
| **Age at First Birth** | | | | | | | | |
| | <18 | 2685 (55.8) | 41.5 | 65.2 | 59.5 | 51.5 | 44.5 | 35.9 |
| | 18–24 | 1865 (38.7) | 57.5 | 31.1 | 36.7 | 41.2 | 47.2 | 51.6 |
| | ≥25 | 264 (5.5) | 82.6 | 3.7 | 3.8 | 7.3 | 8.3 | 12.6 |
| **Mother's BMI** | | | | | | | | |
| | Underweight | 762 (15.8) | 39.8 | 21.9 | 17.5 | 12.7 | 10.8 | 6.6 |
| | Normal | 3005 (62.4) | 46.1 | 68.1 | 70.7 | 60.9 | 60.8 | 44.3 |
| | Overweight/Obese | 1047 (21.7) | 68.4 | 10.0 | 11.8 | 26.4 | 28.3 | 49.1 |
| **Mother's Education** | | | | | | | | |
| | No education | 312 (6.5) | 3.2 | 7.4 | 4.3 | 4.3 | 2.2 | 1.7 |
| | Primary | 1337 (27.8) | 17.8 | 39.6 | 27.1 | 16.1 | 15.1 | 7.3 |
| | Secondary | 2308 (47.9) | 50.8 | 45.6 | 58.4 | 60.2 | 53.8 | 44.8 |
| | Higher | 857 (17.8) | 28.3 | 7.4 | 10.2 | 19.5 | 28.9 | 46.2 |
| **Mother's employment status** | | | | | | | | |
| | Not Working | 3046 (63.3) | 55.2 | 52.6 | 60.1 | 67.8 | 71.3 | 82.5 |
| | Working | 1768 (36.7) | 40.9 | 47.4 | 39.9 | 32.2 | 28.7 | 17.5 |
| **Number of ANC Visits** | | | | | | | | |
| | Nil | 570 (11.8) | 23.2 | 6.7 | 9.4 | 5.4 | 4.6 | 3.9 |
| | 1–3 visit | 2009 (41.7) | 39.6 | 50.4 | 46.4 | 39.1 | 34.3 | 19.0 |
| | ≥ 4 visits | 2235 (46.4) | 66.1 | 43.0 | 44.2 | 55.6 | 61.1 | 77.1 |
| **Husband's Education** | | | | | | | | |
| | No education | 676 (14.0) | 29.4 | 23.0 | 13.9 | 8.8 | 5.6 | 2.0 |
| | Primary | 1637 (34.0) | 38.5 | 46.7 | 43.2 | 27.5 | 23.2 | 10.6 |
| | Secondary | 1592 (33.1) | 54.1 | 22.6 | 30.6 | 42.3 | 44.7 | 36.8 |
| | Higher | 909 (18.9) | 78.5 | 7.8 | 12.3 | 21.5 | 26.4 | 50.6 |
| **Husband's Occupation** | | | | | | | | |
| | Agricultural | 911 (18.9) | 33.5 | 31.1 | 25.5 | 16.7 | 9.0 | 2.3 |
| | Business | 1051 (21.8) | 56.4 | 8.5 | 14.7 | 23.6 | 26.0 | 34.7 |
| | Job/Services | 2268 (47.1) | 57.1 | 38.1 | 48.8 | 52.1 | 57.0 | 58.4 |
| | Others | 584 (12.1) | 36.3 | 22.2 | 11.0 | 7.5 | 8.0 | 4.6 |
| **Mother's Empowerment** | | | | | | | | |

*(Continued)*

**Table 2.** (Continued)

|  |  | Total (n/%) | Facility birth (%) | Poorest | Poorer | Middle | Richer | Richest |
|---|---|---|---|---|---|---|---|---|
| **Decision-Making Power** |  |  |  |  |  |  |  |  |
|  | Low | 2625 (54.5) | 50.6 | 27.0 | 56.7 | 52.8 | 51.1 | 54.9 |
|  | High | 2189 (45.5) | 49.2 | 26.5 | 43.3 | 47.2 | 48.9 | 45.1 |
| **Mother's Enlightenment** |  |  |  |  |  |  |  |  |
|  | Low | 1870 (38.8) | 31.3 | 61.1 | 66.3 | 44.4 | 21.5 | 13.4 |
|  | Medium | 818 (17.0) | 46.3 | 29.6 | 18.1 | 17.7 | 21.0 | 17.5 |
|  | High | 2126 (44.2) | 67.7 | 12.7 | 15.6 | 37.9 | 57.5 | 69.1 |
| **Wife Beating** |  |  |  |  |  |  |  |  |
|  | Low | 3946 (82.0) | 51.7 | 26.3 | 83.3 | 81.2 | 80.7 | 84.2 |
|  | High | 868 (18.0) | 42.1 | 29.5 | 16.7 | 18.8 | 19.3 | 15.8 |

power, were found insignificantly associated with the utilization of facility delivery variable at p<0.05 level of significance.

The results from regression analysis presented in (Table 4) show that urban women in Bangladesh were 1.18 times (AOR = 1.18, 95% CI = 1.01–1.37) more likely to utilize facility delivery than rural women. Facility delivery was found (AOR = 3.40, 95% CI = 2.68–4.31) 3.40 times higher for those who received ≥4 ANC visits, compared to those who didn't seek any ANC during their last pregnancy. With the increase of husband's education, the odds of utilizing facility delivery were also seen to be higher compared to husbands with no education. It was observed that the household wealth status was positively associated with utilizing facility delivery. Middle-class, richer and richest class women were found to be more likely to utilize facility delivery than women from the poorest households. Women with high enlightenment were observed to utilize (AOR = 1.56, 95% CI = 1.32–1.85) 1.56 times higher facility delivery than women with low enlightenment.

Table 5 shows the decomposition of CIX in respect to wealth status. Decomposition of CIX was done to see the contribution of different explanatory variables on the inequalities. The column percentage of contribution in both the table represents the relative contribution of each explanatory variables on the overall CIX. A variable's contribution is expressed as a percentage, with a negative percentage indicating that it helps to reduce concentration and a positive percentage indicating that it helps to raise the observed inequality. In the observed CIX due to wealth status which is 0.205, 57.399% contributed by wealth status, 7.134% contributed by education level of mother, 10.242% contributed by the age at first birth, 8.960% contributed by mother's partner education level, 6.149% contributed by mother's enlightenment.

## Results from the measures of inequality

The average facility delivery per household wealth category is shown in Fig 1. An estimated one-quarter (26.44% 95% CI: 23.83–29.24) of women in the lowest-wealth quintile gave birth in a health facility, compared to nearly one-third of those who were the richest (79.33% of those who gave birth in hospitals, 95% CI: 76.53–81.88). The graph also shows that the level of facility delivery steadily rises in women as one move from the poorest to the richest quintile. While Fig 2 illustrates the inequalities in facility delivery based on one's wealth status. Due to the fact that the concentration curve is below the line of equality, therefore, it is decided that facility delivery was disproportionately higher among women from affluent groups. The relative CIX value for facility delivery is shown in Table 5. The CIX regarding wealth status was 0.205 (p<0.001) indicates the significance of CIX. A positive estimated CIX in suggests that

**Table 3. Bivariate association test between facility delivery and explanatory variables.**

| | | | Utilization of facility birth | | | |
|---|---|---|---|---|---|---|
| | | Total (n/%) | Facility birth | Home birth | P-value | Prevalence (95% CI) |
| **Divisions** | | | | | <0.001 | |
| | Barishal | 518 (10.8) | 41.9 | 58.1 | | 39.32(33.75–45.17) |
| | Chattogram | 796 (16.5) | 46.5 | 53.5 | | 45.49(42.46–48.55) |
| | Dhaka | 672 (14.0) | 60.3 | 39.7 | | 58.27(55.43–61.05) |
| | Khulna | 495 (10.3) | 62.2 | 37.8 | | 60.03(55.38–64.51) |
| | Mymensingh | 589 (12.2) | 42.4 | 57.6 | | 39.14(34.57–43.91) |
| | Rajshahi | 497 (10.3) | 55.9 | 44.1 | | 53.71(49.57–57.81) |
| | Rangpur | 550 (11.4) | 52.0 | 48.2 | | 48.5(44.26–52.77) |
| | Sylhet | 697 (14.5) | 41.8 | 58.2 | | 38.0(33.33–42.90) |
| **Place of Residence** | | | | | <0.001 | |
| | Urban | 1610 (33.4) | 62.5 | 37.5 | | 63.37(60.65–66.01) |
| | Rural | 3204 (66.6) | 43.6 | 56.4 | | 44.99(43.36–46.62) |
| **Age of the mother** | | | | | 0.106 | |
| | 15–24 | 2563 (53.2) | 51.1 | 48.9 | | 51.42(49.48–53.34) |
| | 25–34 | 1974 (41.0) | 49.1 | 50.9 | | 48.48(46.29–50.69) |
| | 35–49 | 277 (5.8) | 45.1 | 54.9 | | 42.08(36.20–48.18) |
| **Age at First Birth** | | | | | <0.001 | |
| | <18 | 2685 (55.8) | 41.5 | 58.5 | | 41.80(39.97–43.66) |
| | 18–24 | 1865 (38.7) | 57.5 | 42.5 | | 57.53(55.25–59.78) |
| | ≥25 | 264 (5.5) | 82.6 | 17.4 | | 80.67(75.19–85.18) |
| **Mother's BMI** | | | | | <0.001 | |
| | Underweight | 762 (15.8) | 39.8 | 60.2 | | 40.27(36.76–43.88) |
| | Normal | 3005 (62.4) | 46.1 | 53.9 | | 46.02(44.26–47.80) |
| | Overweight/Obese | 1047 (21.7) | 68.4 | 31.6 | | 66.87(63.96–69.64) |
| **Mother's Education** | | | | | <0.001 | |
| | No education | 312 (6.5) | 24.4 | 75.6 | | 26.01(21.44–317) |
| | Primary | 1337 (27.8) | 31.9 | 68.1 | | 31.99(29.53–34.56) |
| | Secondary | 2308 (47.9) | 52.9 | 47.1 | | 52.81(50.80-54-82) |
| | Higher | 857 (17.8) | 79.5 | 20.5 | | 78.41(75.45–81.10) |
| **Mother's Employment Status** | | | | | <0.001 | |
| | Not Working | 3046 (63.3) | 55.2 | 44.8 | | 54.86(53.09–56.62) |
| | Working | 1768 (36.7) | 40.9 | 59.1 | | 40.76(38.49–43.07) |
| **Number of ANC Visits** | | | | | <0.001 | |
| | Nil | 570 (11.8) | 23.2 | 76.8 | | 23.53(20.21–27.21) |
| | 1–3 visit | 2009 (41.7) | 39.6 | 60.4 | | 40.07(37.97–42.20) |
| | ≥ 4 visits | 2235 (46.4) | 66.1 | 33.9 | | 65.46(63.45–67.42) |
| **Husband's Education** | | | | | <0.001 | |
| | No education | 676 (14) | 29.4 | 70.6 | | 30.03(26.68–33.61) |
| | Primary | 1637 (34) | 38.5 | 61.5 | | 38.05(35.73–40.42) |
| | Secondary | 1592 (33.1) | 54.1 | 45.9 | | 54.61(52.19–57.01) |
| | Higher | 909 (18.9) | 78.5 | 21.5 | | 77.69(74.80–80.33) |
| **Husband's Occupation** | | | | | <0.001 | |
| | Agricultural | 911 (18.9) | 33.5 | 66.5 | | 34.48(31.52–37.57) |
| | Business | 1051 (21.8) | 56.4 | 43.6 | | 57.41(54.35–60.42) |
| | Job/Services | 2268 (47.1) | 57.1 | 42.9 | | 56.44(54.39–58.48) |
| | Others | 584 (12.1) | 36.3 | 63.7 | | 35.42(31.70–39.31) |

*(Continued)*

**Table 3.** (Continued)

| | | Utilization of facility birth | | | |
|---|---|---|---|---|---|
| | Total (n/%) | Facility birth | Home birth | P-value | Prevalence (95% CI) |
| **Household Wealth Status** | | | | <0.001 | |
| Poorest | 1061(22.04) | 11.35 | 32.71 | | 26.44(23.83–29.24) |
| Poorer | 996(20.69) | 15.34 | 26.03 | | 37.04(34.11–40.07) |
| Middle | 875(18.18) | 18.21 | 18.14 | | 49.92(46.72–53.12) |
| Richer | 958(19.90) | 24.03 | 15.77 | | 60.06(56.96–63.08) |
| Richest | 924(19.19) | 31.06 | 7.35 | | 79.33(76.53–81.88) |
| **Mother's Empowerment** | | | | | |
| **Decision-Making Power** | | | | 0.311 | |
| Low | 2625 (54.5) | 50.6 | 49.4 | | 49.6(47.69–51.520) |
| High | 2189 (45.5) | 49.2 | 50.8 | | 49.83(47.75–51.91) |
| **Mother's Enlightenment** | | | | <0.001 | |
| Low | 1870 (38.8) | 31.3 | 68.7 | | 30.87(28.79–33.03) |
| Medium | 818 (17.0) | 46.3 | 53.7 | | 46.80(43.42–50.21) |
| High | 2126 (44.2) | 67.7 | 32.3 | | 66.67(64.66–68.63) |
| **Wife Beating** | | | | <0.001 | |
| Low | 3946 (82.0) | 51.7 | 48.3 | | 50.93(49.37–52.49) |
| High | 868 (18.0) | 42.1 | 57.9 | | 44.24(40.99–47.54) |

facility delivery was more concentrated among wealthy women than among poorer women [24].

## Discussion

Using nationally representative surveys in Bangladesh, this study found several socioeconomic inequalities in utilizing facility delivery. According to the findings of our analysis, women with a higher level of education are more likely to utilize facility delivery while women with no education are less likely to do so. This connection between education and facility delivery seems obvious, as educated people are more aware of personal health issues, have stronger self-efficacy, and adhere to self-care [25].

Besides educational attainment, our study also found that women in urban areas of Bangladesh are more likely to utilize facility delivery than women living in the rural areas. The plausible explanations could be that urban women are more likely to be educated and as a result, are more health-conscious and have better access to health care services than rural women. This would result in a rise of ANC visits which could further facilitate institutional deliveries. In contrast, rural women were more likely to birth at home owing to their lower socioeconomic status and poor number of ANC visits. However, due to governments and different stakeholders multiple measures including financial and other incentives as well as the construction of rural birth centers, inequalities in the use of facility delivery is decreasing noticeably [17,26,27]. We also found a significant influence of household wealth status on socioeconomic inequalities in utilizing facility delivery where rich and middle-class women are more likely to utilize facility delivery than those from the poorest socioeconomic class. Poverty could be a strong predictor of population health status, and its impact on women's general and reproductive health outcomes is much more pronounced.

It might the fact that low-income households usually spend the lion's share of their budget on food resulting in difficult trade-offs regarding spending on education and health care. The poor class and even those in the middle-income bracket cannot afford the cost of delivery at

**Table 4. Regression results factors associated with facility birth: BDHS 2017–18.**

| Divisions | | UOR (95% CI) | P-Value | AOR (95% CI) | P-Value |
|---|---|---|---|---|---|
| | Barishal (RC) | 1.00 | | 1.00 | |
| | Chattogram | 1.20 (0.96–1.51) | 0.102 | 0.95 (0.73–1.23) | 0.681 |
| | Dhaka | 2.10 (1.67–2.66) | <0.001 | 1.26 (0.95–1.65) | 0.103 |
| | Khulna | 2.28 (1.78–2.94) | <0.001 | 1.78 (1.33–2.37) | <0.001 |
| | Mymensingh | 1.02 (0.81–1.30) | 0.853 | 1.05 (0.80–1.39) | 0.713 |
| | Rajshahi | 1.76 (1.37–2.26) | <0.001 | 1.63 (1.22–2.16) | 0.001 |
| | Rangpur | 1.50 (1.18–1.91) | 0.001 | 1.55 (1.17–2.06) | 0.002 |
| | Sylhet | 0.99 (0.79–1.25) | 0.961 | 1.02 (0.78–1.33) | 0.911 |
| **Place of Residence** | | | | | |
| | Urban | 2.16 (1.91–2.44) | <0.001 | 1.18 (1.01–1.37) | 0.037 |
| | Rural (RC) | 1.00 | | 1.00 | |
| **Age of the mother** | | | | | |
| | 15-24(RC) | 1.00 | | 1.00 | |
| | 25–34 | 0.92 (0.82–1.04) | 0.188 | 0.75 (0.65–0.86) | <0.001 |
| | 35–49 | 0.79 (0.61–1.01) | 0.059 | 0.68 (0.49–0.92) | 0.014 |
| **Age at First Birth** | | | | | |
| | <18 (RC) | 1.00 | | 1.00 | |
| | 18–24 | 1.9 (1.69–2.15) | <0.001 | 1.42 (1.24–1.64) | <0.001 |
| | ≥25 | 6.67 (4.81–9.26) | <0.001 | 3.82 (2.62–5.59) | <0.001 |
| **Mother's BMI** | | | | | |
| | Underweight (RC) | 1.00 | | 1.00 | |
| | Normal | 1.30 (1.10–1.52) | 0.002 | 1.04 (0.86–1.25) | 0.690 |
| | Overweight/Obese | 3.28 (2.70–3.98) | <0.001 | 1.77 (1.41–2.22) | <0.001 |
| **Mother's Education** | | | | | |
| | No education | 1.00 | | 1.00 | |
| | Primary | 1.46 (1.10–1.93) | 0.009 | 1.06 (0.77–1.46) | 0.733 |
| | Secondary | 3.49 (2.66–4.57) | <0.001 | 1.50 (1.07–2.09) | 0.016 |
| | Higher | 12.02 (8.84–16.33) | <0.001 | 2.16 (1.44–3.23) | <0.001 |
| **Mother's employment status** | | | | | |
| | Not Working (RC) | 1.00 | | 1.00 | |
| | Working | 0.56 (0.50–0.63) | <0.001 | 0.67 (0.58–0.77) | <0.001 |
| **Number of ANC Visits** | | | | | |
| | Nil (RC) | 1.00 | | 1.00 | |
| | 1–3 visit | 2.17 (1.75–2.69) | <0.001 | 1.71 (1.36–2.17) | <0.001 |
| | ≥ 4 visits | 6.48 (5.23–8.02) | <0.001 | 3.40 (2.68–4.31) | <0.001 |
| **Husband's Education** | | | | | |
| | No education (RC) | 1.00 | | 1.00 | |
| | Primary | 1.50 (1.24–1.82) | <0.001 | 1.14 (0.92–1.41) | 0.229 |
| | Secondary | 2.83 (2.33–3.43) | <0.001 | 1.32 (1.06–1.65) | 0.014 |
| | Higher | 8.78 (6.98–11.04) | <0.001 | 2.23 (1.69–2.92) | <0.001 |
| **Husband's Occupation** | | | | | |
| | Agricultural | 0.88 (0.71–1.10) | 0.263 | 0.96 (0.75–1.22) | 0.741 |
| | Business | 2.27 (1.85–2.80) | <0.001 | 1.17 (0.92–1.48) | 0.199 |
| | Job/Services | 2.34 (1.94–2.82) | <0.001 | 1.21 (0.97–1.49) | 0.087 |
| | Others (RC) | 1.00 | | 1.00 | |
| **Household Wealth Status** | | | | | |
| | Poorest (RC) | 1.00 | | 1.00 | |

(*Continued*)

**Table 4.** (Continued)

| Divisions | | UOR (95% CI) | P-Value | AOR (95% CI) | P-Value |
|---|---|---|---|---|---|
| | Poorer | 1.70(1.41–2.05) | <0.001 | 1.28(1.04–1.57) | 0.021 |
| | Middle | 2.89(2.39–3.50) | <0.001 | 1.61(1.29–2.01) | <0.001 |
| | Richer | 4.39(3.63–3.30) | <0.001 | 2.00(1.58–2.53) | <0.001 |
| | Richest | 12.18(9.84–15.05) | <0.001 | 3.87(2.54–4.41) | <0.001 |
| **Decision Making Power** | | | | | |
| | Low (RC) | 1.00 | | Not Adjusted in the Final Model | |
| | High | 0.94 (0.84–1.06) | 0.309 | | |
| **Mother's Enlightenment** | | | | | |
| | Low (RC) | 1.00 | | 1.00 | |
| | Medium | 1.89 (1.60–2.24) | <0.001 | 1.11 (0.92–1.34) | 0.273 |
| | High | 4.60 (4.02–5.26) | <0.001 | 1.56 (1.32–1.85) | <0.001 |
| **Wife Beating** | | | | | |
| | Low | 1.47 (1.27–1.71) | <0.001 | 1.07 (0.9–1.27) | 0.442 |
| | High (RC) | 1.00 | | 1.00 | |

RC stands for Reference Category, ANC for Antenatal Care, BMI means to Body Mass Index, Divisions means Administrative Regions.

the government hospitals let alone at private clinic. Consequently, women are compelled to give birth at home with the help of a Traditional Birth Attendant(TBA) [28]. Specifically, it was found that women whose husbands were employed in job or any service had a higher likelihood of delivering at a health facility than women whose husbands were employed in any other professions. As occupation type is related to both education and income, it is a useful predictor of financial access to essential healthcare treatments.

Therefore, national policies targeting to income generation and poverty reduction are likely to have a positive effect on improving the utilization of facility delivery in Bangladesh. Decision-making power is an important factor for maternal health related issues which align with the results of some previous studies [29,30] where it was found that husband's decision-making power was a strong predictor influencing health facility delivery in Pakistan. Men, particularly in South Asian societies, believe that pregnancy and childbirth are solely the responsibility of women and so, they are not actively involved in such issues [10,31]. This is mostly because of the existing cultural differences, religious beliefs, personal views that may result in unequal in decision-making authority in this region [32,33].

However, according to several studies, joint choice, or effective communication between partners increases facility delivery usage more than individual decisions [34]. Results revealed that highly enlightened mothers are more likely to use facility delivery than less educated mothers. Other studies also described how educational status influences care seeking from hospitals [35–37]. Particularly women with more education are more likely to go for facility delivery. In contrast to past research in which working women more self-sufficient and hence had greater decision-making power over reproductive care and ANC visits [10,34] that led to facility delivery, working women were again less likely to use a facility delivery. Additionally, the working women also enjoy greater financial stability, which enables them to handle the costs of facility delivery. Therefore, furthering facility delivery and improving women's health requires more than just education; concerns of women's empowerment and economic security must also be addressed [11,19].

Due to the fact that only 39.7% of deliveries take place in hospitals in Bangladesh [38], underutilization of hospital-based interventions is unavoidable. Again, in Bangladesh, the

**Table 5. Decomposition of concentration index for measuring socioeconomic inequalities in utilizing facility delivery in Bangladesh.**

| Variables | | Elasticity | CIX | Contribution to overall CIX = 0.205 (p<0.001) | |
|---|---|---|---|---|---|
| | | | | **Absolute contribution** | **Percentage contribution** |
| **Divisions** | | | | | |
| | Barishal (RC) | | | | |
| | Chattogram | -0.012 | 0.075 | -0.001 | -0.450 |
| | Dhaka | 0.045 | 0.259 | 0.009 | 3.630 |
| | Khulna | 0.041 | 0.036 | 0.001 | 0.717 |
| | Mymensingh | -0.003 | -0.204 | 0.001 | 0.264 |
| | Rajshahi | 0.041 | -0.072 | -0.003 | -1.419 |
| | Rangpur | 0.035 | -0.302 | -0.011 | -6.202 |
| | Sylhet | 0.001 | -0.111 | -0.001 | -0.005 |
| | Subtotal | | | **-0.005** | **-3.465** |
| **Place of Residence** | | | | | |
| | Rural (RC) | | | | |
| | Urban | 0.029 | 0.376 | 0.011 | 4.246 |
| | Subtotal | | | **0.011** | **4.246** |
| **Age of the Mother** | | | | | |
| | 15–24 years (RC) | | | | |
| | 25–34 years | -0.083 | 0.022 | -0.002 | -0.880 |
| | 35–49 years | -0.015 | -0.008 | 0.001 | 0.056 |
| | Subtotal | | | **-0.001** | **-0.824** |
| **Mother's Education** | | | | | |
| | No education (RC) | | | | |
| | Primary | -0.001 | -0.259 | 0.002 | 0.906 |
| | Secondary | 0.110 | 0.040 | 0.004 | 2.163 |
| | Higher | 0.078 | 0.420 | 0.013 | 4.065 |
| | Subtotal | | | **0.019** | **7.134** |
| **Age at First Birth** | | | | | |
| | <18 years (RC) | | | | |
| | 18–24 years | 0.086 | 0.114 | 0.010 | 4.806 |
| | ≥ 25 years | 0.044 | 0.343 | 0.015 | 5.436 |
| | Subtotal | | | **0.025** | **10.242** |
| **Mother's BMI** | | | | | |
| | Normal (RC) | | | | |
| | Underweight | -0.006 | -0.195 | 0.001 | 0.546 |
| | Overweight/Obese | 0.062 | 0.293 | 0.012 | 5.628 |
| | Subtotal | | | **0.013** | **6.174** |
| **Mother's Employment** | | | | | |
| | Not working (RC) | | | | |
| | Working | -0.097 | -0.171 | 0.017 | 5.096 |
| | Subtotal | | | **0.017** | **5.096** |
| **Number of ANC Visits** | | | | | |
| | Nil (RC) | | | | |
| | 1–3 visit | 0.188 | -0.103 | -0.019 | -9.502 |
| | ≥ 4 visits | 0.278 | 0.137 | 0.029 | 7.622 |
| | Subtotal | | | **0.010** | **-1.880** |
| **Husband's Education** | | | | | |

*(Continued)*

**Table 5.** (Continued)

| Variables | | Elasticity | CIX | Contribution to overall CIX = 0.205 (p<0.001) | |
|---|---|---|---|---|---|
| | | | | **Absolute contribution** | **Percentage contribution** |
| | No education (RC) | | | | |
| | Primary | 0.008 | -0.199 | -0.002 | -0.803 |
| | Secondary | 0.041 | 0.123 | 0.005 | 2.456 |
| | Higher | 0.070 | 0.420 | 0.019 | 7.307 |
| | Subtotal | | | **0.022** | **8.960** |
| **Husband's Occupation** | | | | | |
| | Others (RC) | | | | |
| | Agricultural | 0.004 | -0.332 | -0.001 | -0.686 |
| | Business | 0.036 | 0.204 | 0.007 | 3.609 |
| | Job/Services | 0.083 | 0.109 | 0.009 | 4.429 |
| | Subtotal | | | **0.015** | **7.352** |
| **Household Wealth Status** | | | | | |
| | Poorest (RC) | | | | |
| | Poor | 0.031 | -0.368 | -0.011 | -5.561 |
| | Middle | 0.058 | 0.034 | 0.002 | 0.963 |
| | Rich | 0.095 | 0.431 | 0.035 | 13.028 |
| | Richest | 0.148 | 0.817 | 0.103 | 48.969 |
| | Subtotal | | | **0.129** | **57.399** |
| **Mother's Empowerment** | | | | | |
| **Decision-making Power** | | | | | |
| | Low (RC) | | | | |
| | High | -0.028 | 0.020 | -0.001 | -0.279 |
| | Subtotal | | | **-0.001** | **-0.279** |
| **Mother's Enlightenment** | | | | | |
| | Low (RC) | | | | |
| | Medium | 0.021 | 0.014 | 0.001 | 0.139 |
| | High | 0.095 | 0.306 | 0.019 | 6.010 |
| | Subtotal | | | **0.020** | **6.149** |
| **Wife Beating** | | | | | |
| | High (RC) | | | | |
| | Low | -0.015 | 0.019 | -0.001 | -0.142 |
| | Subtotal | | | **-0.001** | **-0.142** |
| **Explained CI** | | | | **0.273** | **106.162** |
| **Residual CI** | | | | **-0.068** | **-6.162** |

CI: Concentration Index.

most prevalent reasons for preferring home birth with a TBA include traditional attitudes, religious misconceptions, restricted decision-making power of women within family and difficulty in reaching the nearest health facility [39]. In addition, many people prefer home delivery due to lack of understanding regarding service delivery points, a fear of caesarean delivery at hospitals, and a dearth of female doctors in healthcare facilities [39]. The facilities in high-end health sector are not sufficient. Though staff training is required to improve the technical competence of health service providers, in-service trainings are relatively ineffective. Therefore, additional investment in human resources is required [40,41].

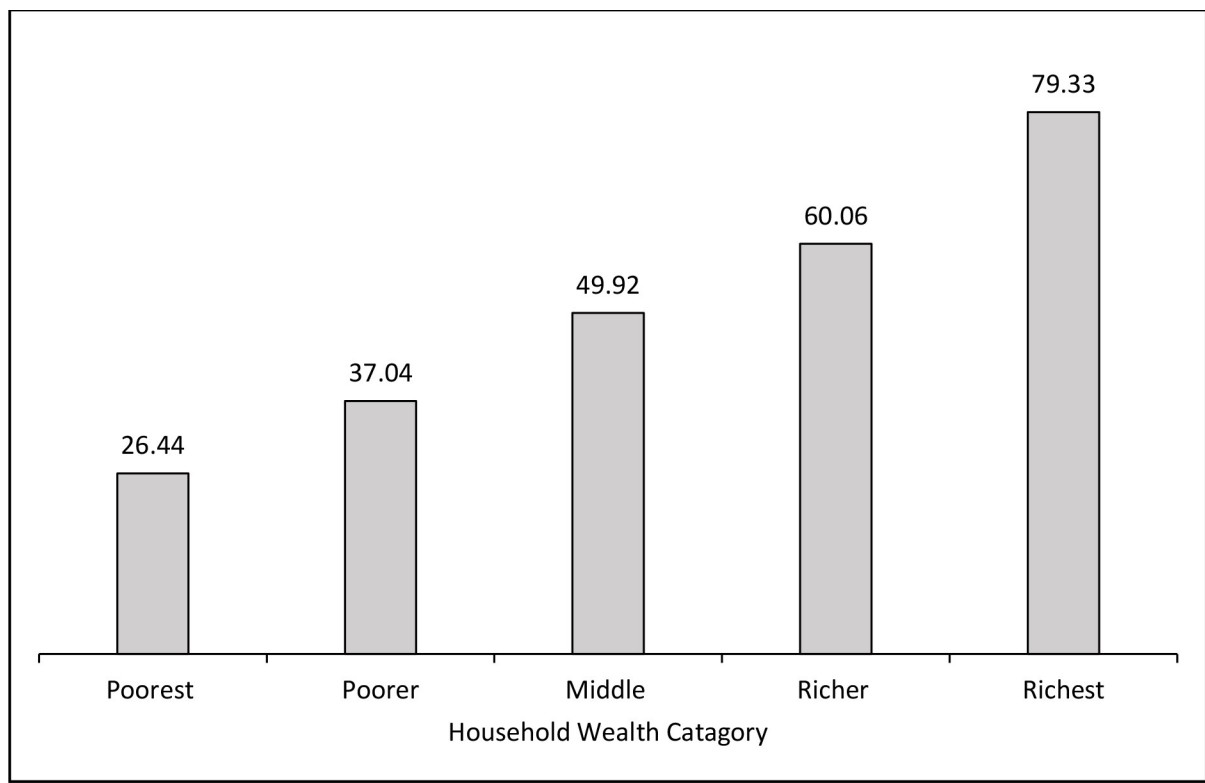

**Fig 1. Facility delivery by household wealth rank.**

This study's strengths and limitations were acknowledged with caution. Firstly, this study used the recent round nationally representative survey data therefore the findings could be generalizable to the entire population of Bangladesh. Secondly, standard statistical methods have been employed to estimate the prevalence and the decomposition of inequality measures. The inherent limitations of a cross-sectional study design limited our ability to infer causality. Some potential factors such as distance to the nearest institution with a birthing facility, cost (direct or indirect) of facility delivery, waiting time, behavior of the healthcare practitioners, and the availability of transportation facilities and awareness of the importance of safe delivery were excluded from this analysis because they were not included in the original DHS data.

## Conclusions

Even though the rate if facility delivery births in Bangladesh is low, a considerable percentage of women continue to give birth at home. There are inequalities in the utilization of facility delivery exist in urban and rural areas in Bangladesh, as well as between the wealth class of households. The findings of the study demand the revision of strategies and programs to eliminate the disparities and improve facility delivery service utilization. The factors that are closely correlated to utilizing the facility delivery are educational attainment, number of ANC visits, place of residence, women's ability to make health-related decisions. Based on these insights, the health policymakers need to consider implementing special intervention programs aiming to reduce inequality and improve the utilization of facility delivery in Bangladesh. This would tremendously contribute to achieving Sustainable Development Goals (SDGs) in line with the government's own visions.

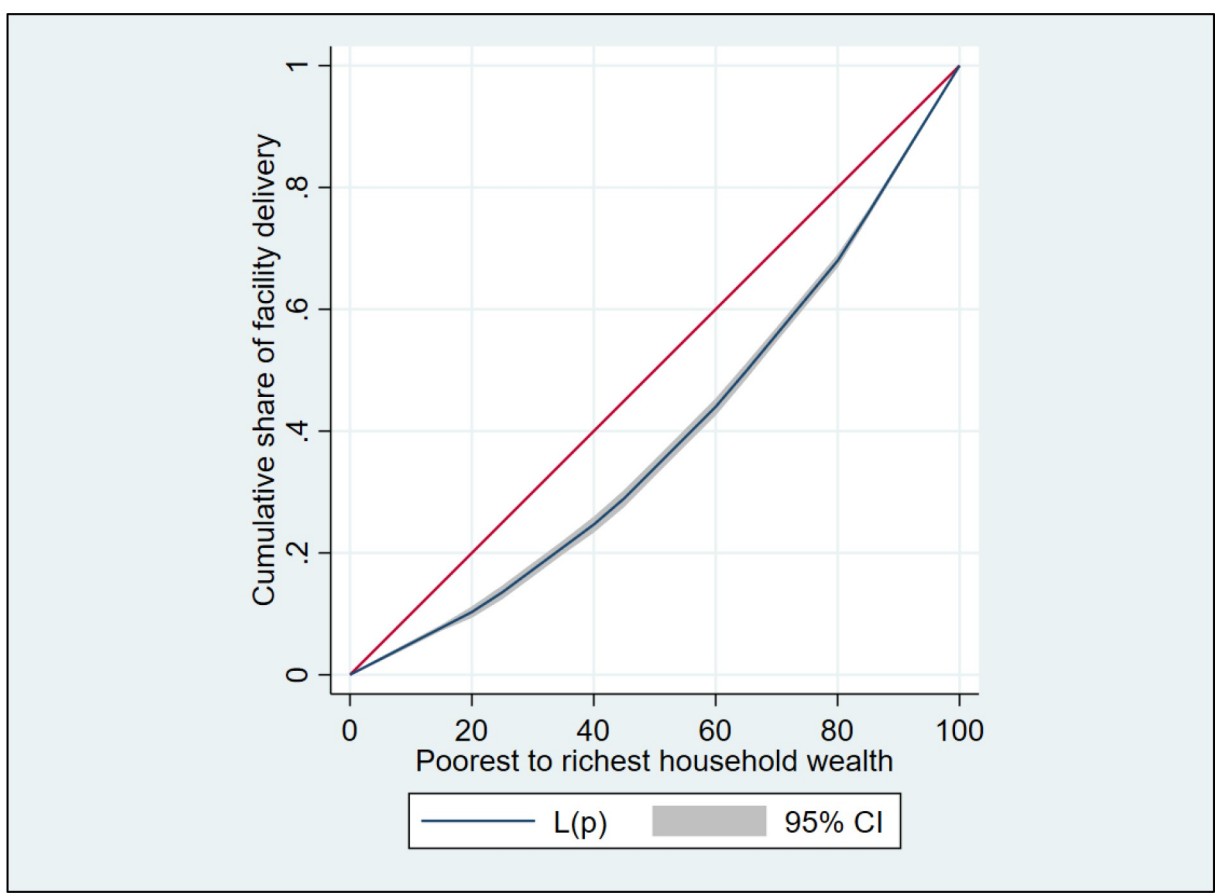

**Fig 2. Concentration curve for facility delivery against household wealth category.**

## Acknowledgments

The authors are grateful to the Demographic and Health Surveys (DHS) Program for providing BDHS data accessibility for conducting the study. The authors also want to give a big thank to Umesh Prasad Bhusal (https://orcid.org/0000-0001-9331-6028) for his outstanding help for guiding us to draw concentration curve.

## Author Contributions

**Conceptualization:** Md. Ashfikur Rahman.

**Data curation:** Md. Ashfikur Rahman.

**Formal analysis:** Md. Ashfikur Rahman, Satyajit Kundu, Md. Mahmudur Rahman.

**Investigation:** Md. Ashfikur Rahman, Md. Mahmudur Rahman.

**Methodology:** Md. Ashfikur Rahman, Satyajit Kundu.

**Supervision:** Md. Ashfikur Rahman.

**Validation:** Md. Ashfikur Rahman, Md. Hasan Howlader.

**Visualization:** Md. Ashfikur Rahman.

**Writing – original draft:** Md. Ashfikur Rahman, Satyajit Kundu, Harun Or Rashid, Hasibul Hasan Shanto, Md. Mahmudur Rahman, Bayezid Khan, Md. Hasan Howlader, Md. Akhtarul Islam.

**Writing – review & editing:** Md. Ashfikur Rahman, Satyajit Kundu, Harun Or Rashid, Hasibul Hasan Shanto, Md. Mahmudur Rahman, Bayezid Khan, Md. Hasan Howlader, Md. Akhtarul Islam.

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
