## [Decision Letter · Decision Letter 0]

25 Jul 2022

PONE-D-22-13608Socioeconomic Inequalities in Utilizing Facility Delivery in Bangladesh: A Decomposition Analysis Using Nationwide 2017-2018 Demographic and Health Survey DataPLOS ONE

Dear Dr. Rahman,

Thank you for submitting your manuscript to PLOS ONE. After careful consideration, we feel that it has merit but does not fully meet PLOS ONE’s publication criteria as it currently stands. Therefore, we invite you to submit a revised version of the manuscript that addresses the points raised during the review process.

We look forward to receiving your revised manuscript.

Kind regards,

Jayanta Kumar Bora, PhD

Academic Editor

PLOS ONE

Journal Requirements:

- https://journals.plos.org/plosone/article?id=10.1371%2Fjournal.pone.0250012

In your revision ensure you cite all your sources (including your own works), and quote or rephrase any duplicated text outside the methods section. Further consideration is dependent on these concerns being addressed

Reviewers' comments:

Reviewer's Responses to Questions

**Comments to the Author**

1. Is the manuscript technically sound, and do the data support the conclusions?

Reviewer #1: Yes

Reviewer #2: Yes

2. Has the statistical analysis been performed appropriately and rigorously? 

Reviewer #1: Yes

Reviewer #2: Yes

3. Have the authors made all data underlying the findings in their manuscript fully available?

Reviewer #1: Yes

Reviewer #2: Yes

4. Is the manuscript presented in an intelligible fashion and written in standard English?

Reviewer #1: No

Reviewer #2: No

5. Review Comments to the Author

Reviewer #1: Socioeconomic Inequalities in Utilizing Facility Delivery in Bangladesh: A Decomposition Analysis Using Nationwide 2017-2018 Demographic and Health Survey Data.

The authors have chosen appropriate topic for the study as high maternal mortality is one of the challenge before achieving sustainable development. This can be reduced by pushing maternal healthcare and one among them is institutional delivery/facility based delivery. Authors’ efforts are highly appreciated but there are some suggestions for the improvement of the quality of the paper along with expectation about newness in the paper-

1. The research study aimed to examine socio-economic inequality in facility delivery in Bangladesh using latest available cross sectional data for 2017/18. It’s fine. But complexity of framework is missing somewhere as authors included demographic to socio-economic, regional to empowerment & autonomy to programme factors. So, broadly we can say demand and supply side factors. Even with the demand side factors, authors should follow appropriate framework and including so many dimensions seems not needed.

2. In the inequality measurement section, the authors have elaborated much about the advantages of CI and CC, as it is well known and accepted measure of inequality along income/wealth score. The authors are advices to compress it.

3. In line number 233/234, ‘The CIX accepts values between one and one’- this should be corrected as ‘The CIX accepts values between minus one and plus one’.

4. Why two categories for decision making power and wife beating whereas three category for mother’s enlightenments.

5. Line number 195-239, not needed too much elaboration as well as repetition. Be concise with more citation and studies pertaining to it.

6. In table 2; check for residence, decision making power, mother entitlement, wife beating etc as they are some mistake when analysing distribution of facility based delivery on wealth index category.

7. In table 3, check for wealth status.

8. It table 4, possibly, authors have missed to add the information about the variable on which the model is adjusted for as the cOR and AOR seems having differential magnitude of relative risk. So, it would be really helpful for the readers, if authors can add about the adjustment made into the logistic regression.

9. “While Figure 2 illustrates the inequalities in facility delivery based on one's wealth status. Due to the fact that the concentration curve is above the line of equality, facility delivery was disproportionately higher among women from affluent groups. The relative CIX value for facility delivery is shown in Table 5. A positive estimated CIX in suggests that facility delivery was more concentrated among wealthy women than among poorer women”- The statement made based on the figure 2 is incorrect. The CC appears above the line of equality, when the outcome of interest is high among the poor. In the figure 2, the CC is below the line of equality. Even in table 5, the sign of CI value is positive. So, be careful when interpreting such inequality measure. So, please do correct it.

10. Authors are advised to interpret the results of table 5 [decomposition of CI]as it is missing in the manuscript.

11. Figure 1 needs appropriate formatting, look it carefully.

12. Line 356-360 pertaining to women work status and place of delivery. In case of women work, type of work regulates more to access to services rather working or not working status. Think the women who are working but are engaged in agricultural activities to those who are in white collar jobs. So, the findings on those lines are also needs corroboration.

13. Discussion section mostly revolved around education and economic status. Authors should have a thought on it and restructure it.

14. In the acknowledgement section, Lorenz curve is typed which we draw in case of Gini based inequality but when we use Concentration Index, the curve is concentration curve.

14. Proper editing is also needed.

Reviewer #2: The authors attempt to study an important issue in the context of developing countries. This study comprehensively analyzed the factors determining the use of facility delivery in Bangladesh. However, before its acceptance, this study needed major revision to make it more apparent to the readers.

The author provides a strong background for the study; however, it needs to add the following points in this section.

1. The authors should emphasize the relevance of this study in the context of Bangladesh.

2. The evidence of the socioeconomic inequality in the utilization of facility delivery from the previous study is missing in the background. The authors needed to provide an explicit summary of what is already known from the earlier studies and the additional contribution of the current research on this topic.

Data sources:

In the data section, the author needed to provide some information about the data, such as the number of households, the number of women interviewed, and the sample size.

The outcome variable study period is missing from the manuscript. The reference period of the outcome variable must be mentioned in detail of the outcome variable, whether facility delivery is analyzed for the last birth or the birth in some specific years.

Explanatory variable:

The justification for using the explanatory variable in the study is missing. Particularly how the mother's age and wife's Beating are associated with the delivery facility's utilization, these variables are also not interpreted in the results and discussion sections. In addition to the controlled variables, the model can control for any complications before the delivery if any information is given in the data.

In the methods section, a few lines are repeated; for example, lines 211-214 and 221-124 are repeated and also needed to make clearer these sentences.

In the equations, details of each component should be cleared to the readers; for example, what's mean is μ.

Results:

The authors used unadjusted logistic regression, but there is no single sentence about the results in the result section. Moreover, the results from the models and decompositions needed to give more interpretation.

6. PLOS authors have the option to publish the peer review history of their article (what does this mean?). If published, this will include your full peer review and any attached files.

Reviewer #1: **Yes: **RAJESH RAUSHAN

Reviewer #2: **Yes: **Moradhvaj

---

## [Author Response · Author response to Decision Letter 0]

9 Aug 2022

All comments have been addressed and provided authors response to reviewer section.

---

## [Decision Letter · Decision Letter 1]

10 Nov 2022

Socioeconomic Inequalities in Utilizing Facility Delivery in Bangladesh: A Decomposition Analysis Using Nationwide 2017-2018 Demographic and Health Survey Data

PONE-D-22-13608R1

Dear Md. Ashfikur Rahman,

We’re pleased to inform you that your manuscript has been judged scientifically suitable for publication and will be formally accepted for publication once it meets all outstanding technical requirements.

Kind regards,

Jayanta Kumar Bora

Academic Editor

PLOS ONE

Additional Editor Comments (optional):

Reviewers' comments:

Reviewer's Responses to Questions

**Comments to the Author**

1. If the authors have adequately addressed your comments raised in a previous round of review and you feel that this manuscript is now acceptable for publication, you may indicate that here to bypass the “Comments to the Author” section, enter your conflict of interest statement in the “Confidential to Editor” section, and submit your "Accept" recommendation.

Reviewer #1: All comments have been addressed

2. Is the manuscript technically sound, and do the data support the conclusions?

Reviewer #1: Yes

3. Has the statistical analysis been performed appropriately and rigorously? 

Reviewer #1: Yes

4. Have the authors made all data underlying the findings in their manuscript fully available?

Reviewer #1: Yes

5. Is the manuscript presented in an intelligible fashion and written in standard English?

Reviewer #1: Yes

6. Review Comments to the Author

Reviewer #1: The authors have addressed all the comments and suggestions. The background to data and method sections have improved with the inputs provided by the reviewers. The discussion section has been also strengthened with the required inputs.Now No other comments at this stage.

7. PLOS authors have the option to publish the peer review history of their article (what does this mean?). If published, this will include your full peer review and any attached files.

Reviewer #1: **Yes: **RAJESH RAUSHAN

---

## [Editor Report · Acceptance letter]

16 Nov 2022

PONE-D-22-13608R1 

Socioeconomic Inequalities in Utilizing Facility Delivery in Bangladesh: A Decomposition Analysis Using Nationwide 2017-2018 Demographic and Health Survey Data 

Dear Dr. Rahman:

I'm pleased to inform you that your manuscript has been deemed suitable for publication in PLOS ONE. Congratulations! Your manuscript is now with our production department. 

Kind regards, 

on behalf of

Dr. Jayanta Kumar Bora 

Academic Editor

PLOS ONE